



# Improved hydrometeor detection near the Earth's surface by a conically scanning spaceborne W-band radar

Marco Coppola[1], Alessandro Battaglia[1,2], Frederic Tridon[1], and Pavlos Kollias[3,4]

[1]DIATI, Politecnico of Torino, Turin, Italy
[2]Department of Physics and Astronomy, University of Leicester, Leicester, UK
[3]School of Marine and Atmospheric Sciences, Stony Brook University, Stony Brook, NY, USA
[4]Department of Atmospheric and Oceanic Sciences, McGill University, Montreal, QC Canada
**Correspondence:** marco_coppola@polito.it

**Abstract.** The Earth's strong radar surface return limits the detection of clouds and precipitation in the lowest part of the atmosphere by nadir-pointing spaceborne radars such as CloudSat and EarthCARE. The strength of the Earth's surface radar return is significantly reduced at non-zero incidence angles. The WIVERN 94 GHz radar, currently undergoing Phase A studies by ESA, employs a 3-meter antenna and conical radar sampling at high incidence angles. Here, the benefits of the narrow field

5  of view and the reduction in the Earth's surface return for studying clouds and precipitation in the lowest kilometers of the atmosphere are quantified. The WIVERN radar is expected to improve the signal (hydrometeors) to clutter (surface return) ratio over ice-free ocean surfaces and marginally worsen it over land and sea ice. The impact of these findings on the detection of light rainfall and snowfall near the Earth's surface is discussed.

## 1 Introduction

Spaceborne radar observations are generally hampered by the Earth's strong surface return (also referred to as "clutter") that tends to obscure the hydrometeor signal for ranges near the ground and to introduce a "blind zone" near the surface that is detrimental to the accurate quantification of surface precipitation (Maahn et al., 2014; Schirmacher et al., 2023) and the detection of shallow clouds (Burns et al. (2016); Lamer et al. (2020)). The vertical extent, strength, and morphology of the

15  "clutter" profile depend on the radar frequency, incidence angle, and the transmitted radar pulse characteristics (i.e., pulse length and modulation Beauchamp et al. (2017)). At heights above the Earth's surface, where the clutter is stronger than the radar receiver's thermal noise, the clutter determines the radar sensitivity near the Earth's surface. Knowing the shape of the clutter profile allows the signal-to-clutter ratio to be determined, and the clutter to be subtracted to retrieve the atmospheric signal. This is typically done over the ocean, where the shape of the clutter profile exhibits low variability, as demonstrated for

20  the CloudSat Cloud Profiling Radar (CPR) (see the Appendix of Tanelli et al. (2008)) and the National Aeronautics and Space



Administration (NASA) and Japan Aerospace Exploration Agency (JAXA) Global Precipitation Measuring (GPM) mission Dual Precipitation Radar (DPR) in Kubota et al. (2016).

Meneghini and Kozu (1990) suggested that high incidence scanning angles (similar to scatterometers) can significantly reduce the blind zone due to the reduced surface normalized radar cross section (NRCS) moving away from the nadir-looking configuration. Based on wide swath test measurements of the NASA-JAXA Tropical Rainfall Measuring Mission (TRMM) Precipitation Radar (PR) and the GPM DPR, Takahashi (2017) showed that, due to the increasing angle of incidence, a swath width almost twice that of the current GPM/PR swath (250 km) results in a clutter profile with a broader shape (mainly determined by the antenna beamwidth, receiver response, and pulse width; Kanemaru et al. (2020)) but with increasingly smaller surface reflectivity, especially over the ocean, where the NRCS decreases sharply with the angle of incidence. In the case of the TRMM PR, at incidence angles of the order of $30°$, only relatively intense precipitation echoes close to the ground can be targeted, but relatively weak and shallow precipitation will be masked by the clutter.

In addition to the main lobe, issues related to the side lobes of the antenna grating also become important when dealing with an electronically scanned slot array antenna, such as that used for the TRMM and GPM PR (Yamamoto et al., 2020).

WIVERN (WInd Velocity Radar Nephoscope, www.wivern.polito.it; Illingworth et al. (2018); Battaglia et al. (2022)), a novel concept of a wide-swath scanning W-band radar, was proposed in 2020 as part of the European Space Agency (ESA) Earth Explorer programme for studying winds within cloud and precipitation systems (Battaglia et al., 2022; Tridon et al., 2023). After two down-selections, WIVERN is now undergoing Phase A studies, with a final down-selection against the competing CAIRT (Charting the Middle Atmosphere in the Climate System) mission (https://www.cairt.eu/) scheduled for July 2025.

An important advantage of this radar concept, compared to its W-band nadir-pointing predecessors— NASA's CloudSat and EarthCARE Cloud Profiling Radars (CPR; Tanelli et al. (2008); Kollias et al. (2023))—is its large swath (approximately 800 km), which enables much better sampling of the vertical structure of clouds and precipitation and their mesoscale and synoptic scale organization. For example, Scarsi et al. (2024) demonstrated that WIVERN could significantly reduce sampling errors in snowfall observations, bringing them well below natural inter-annual variability at regional and monthly scales.

However, a fair comparison between nadir-looking and conically scanning radars must also consider the impact of clutter.

In terms of surface clutter, the WIVERN W-band radar offers several advantages over the GPM-DPR (which represents the only example of spaceborne scanning atmospheric radar):

- a significantly smaller beamwidth (approximately $0.07°$, compared to $0.7°$). As discussed in Kanemaru et al. (2020), the vertical extent of the clutter is determined by the interplay between the antenna beamwidth and the pulse length;

- the use of an elliptical reflector antenna that mechanically rotates, which results in weaker sidelobes compared to an electronically scanned antenna;

- the further reduction of $\sigma_0$ when moving to incidence angles greater than $40°$, as envisaged for WIVERN;

- the smaller wavelength (W vs $K_a$ and $K_u$), which increases the signal-to-clutter ratio due to the $\lambda^4$ dependence in the surface clutter peak reflectivity (Kollias et al., 2007).



It is, therefore, timely to investigate how a conically scanning configuration could be beneficial or detrimental in terms of reducing or increasing the signal-to-clutter ratio for precipitation (both solid and liquid) near the surface.

After introducing the methodology (Sect. 2), examples of the simulations are presented in Sect. 3.1. A statistical analysis is provided in Sect. 3.2, and conclusions along with future work are outlined in Sect. 4.

## 2    Methodology

The WIVERN mission concept orbit and W-band radar technical specifications are listed in Tab. 1. The sampling geometry is illustrated in Fig. 1, with the conically scanning radar observations, with an angle of incidence, $\theta_{inc}$ of about $42°$.

**Table 1.** The WIVERN mission orbit and W-band radar technical specifications, as currently under study in the Phase-A study for the ESA Earth Explorer 11 program by two industrial consortia. When two values are listed they correspond to the two possible options.

| | |
|---|---|
| Radar output frequency | 94.05 GHz |
| Spacecraft height, $h_{SC}$ | 500 km |
| Spacecraft velocity, $v_{SC}$ | 7600 ms$^{-1}$ |
| Orbit inclination, $i$ | 97.42° |
| Orbit Local Time of the ascending node, $LTAN$ | 06:00 |
| Incidence angle, $\theta_{inc}$ | 41.6°-42.5° |
| Swath width at the ground | 800-825 km |
| Antenna angular velocity, $\Omega_a$ | 12-11 rpm |
| Antenna elevation 3 dB beamwidth, $\theta_p$ | 0.0328 deg |
| Antenna azimuth 3 dB beamwidth, $\phi_p$ | 0.0361 deg |
| Footprint speed | ≈500-475 kms$^{-1}$ |
| Pulse width $\tau$ | 3.3 $\mu$s |
| Single pulse minimum detectable reflectivity | -18 dBZ |
| Minimum detectable signal (MDS) @ 1 km integration | -22.5 dBZ |







**Figure 1.** WIVERN geometry of observation with the conical scanning pattern. The width of the footprint is exaggerated for illustration purposes.

## 2.1 Surface clutter profile

The power received by a space-borne radar from the surface at range $r$, $P_r$, is derived by an integration performed over the illuminated area, $S$, as detailed in Meneghini and Kozu (1990):

$$P_r(r) = \underbrace{\left[ P_t \frac{\lambda^2}{(4\pi)^3} G_0^2 \right]}_{C_S} \int_S \frac{\sigma_0(\psi) \, G_n^2 \, |u(t - 2r/c)|^2}{r^4} \, dS \tag{1}$$

where $P_t$ is the transmitted power, $\lambda$ is the wavelength of radar, $G = G_0 G_n$ is the antenna gain ($G_0$ being the maximum gain at antenna boresight whereas $G_n$ is the antenna gain normalised by the condition $\int_{4\pi} G_n^2 d\Omega = 1$), $u(t)$ is the complex voltage envelope of the transmitted pulse (for a top hat shape $|u(t)| = 1$ for $0 < t < \tau_p$), $\psi$ is the local incidence angle. When working





with flat surfaces, the integral extends to an annular strip of terrain (Battaglia et al. (2017)) while in presence of orography the integral must be evaluated numerically as discussed in Manconi et al. (2024). If, in addition to the flat terrain assumption the antenna has a Gaussian pattern, then the shape of the received power at a scanning angle $\theta_s$ in Eq. (1) will be Gaussian and it can be written in analytical form as (Kanemaru et al., 2020):

$$P_r(r) = P_r(r_s) \exp\left(-8 \log(2) \frac{(\theta(r) - \theta_s)^2}{\theta_p^2 + \theta_b^2}\right) \tag{2}$$

where $r_s$ is the range of the surface, $\theta_b$ is the beamwidth the along range direction, $\theta_p$ accounts for the additional beamwidth introduced in the along range direction because of the pulse width (equal to $c\tau_p$ where $c$ is the speed of light in vacuum and $\tau_p$ is the -6 dB width of the received pulse). In a nutshell, the pulse width produces an extra broadening $\theta_p$ that can be expressed as (Kanemaru et al., 2020):

$$\theta_p = \frac{c\tau_p}{2} \frac{1}{r_s \tan\theta_{inc}}. \tag{3}$$

The incidence angle $\theta_{inc}$ is related to $\theta_s$ by the law of sines:

$$\theta_s = \mathrm{asin}\left(\frac{R_{Earth}}{R_{Earth} + H_{sat}} \sin\theta_{inc}\right) \tag{4}$$

where $H_{sat}$ and $R_{Earth}$ are the height of the satellite and the radius of the Earth. $\theta$ is related to the range $r$ by the law of cosine:

$$\theta(r) = \mathrm{acos}\left(\frac{H_{sat}^2 + 2H_{sat}R_{Earth} + r^2}{2(H_{sat} + R_{Earth})r}\right). \tag{5}$$

By combining Eq. (5) with Eq. (2) it is then possible to analytically derive the shape of flat surface return for radars with circular Gaussian antennas and with pulse having top hat shape.

Then, by using the conversion from power to radar reflectivity discussed in Manconi et al. (2024):

$$P_r(r) = C_M \frac{Z(r)}{r^2} \qquad \text{where } C_M \equiv \frac{\pi^2}{2^6} \frac{P_t\, G_0^2 \Omega_{2A}}{\lambda^2} \frac{c\tau_p}{2} |K_w|^2 = C_S \frac{\pi^5 |K_w|^2 \Omega_{2A}}{\lambda^4} \frac{c\tau_p}{2} \tag{6}$$

where $K_w$ is derived from the refractive index of water at 3 mm-wavelengths ($|K_w|^2$ assumed equal to 0.75), $\Omega_{2A} \equiv \int G_n^2 d\Omega$ (which for a Gaussian beam is approximately equal to $\frac{\pi\theta_b\phi_b}{8\log(2)}$) the received reflectivity at the range of $r_s$ can be computed from the received signal power described in Kanemaru et al. (2020) (their formula 1)

$$Z(r_s) = \frac{\lambda^4}{\pi^5 |K_w|^2} \frac{\theta_{pb}/\theta_b}{c\tau_p/2} \frac{\sigma_0}{\cos\theta_{inc}} L_p \tag{7}$$

where $L_p$ is a peak loss factor and

$$\theta_{pb} = \sqrt{\frac{1}{\frac{1}{\theta_p^2} + \frac{1}{\theta_b^2}}} \tag{8}$$

can be regarded as an effective beamwidth at the surface along the range (i.e. cross-track) direction. Note that for WIVERN $\theta_b = 0.0011$ rad, $\theta_p = 0.00085$ rad so that $\theta_{pb} = 0.00068$ rad and $\theta_{pb}/\theta_b = 0.596$.





Note that $L_p$ can be derived by imposing:

$$\int\limits_{-\infty}^{+\infty} Z(r)dr = \int\limits_{-\infty}^{+\infty} Z(r_s) \exp\left(-8\,\log(2)\,\frac{(\theta-\theta_s)^2}{\theta_p^2+\theta_b^2}\right)dr = \frac{\lambda^4}{\pi^5|K_w|^2}\frac{\sigma_0}{\cos\theta_{inc}} \tag{9}$$

## 2.2 Shape of clutter reflectivity profile for CloudSat, EarthCARE and WIVERN

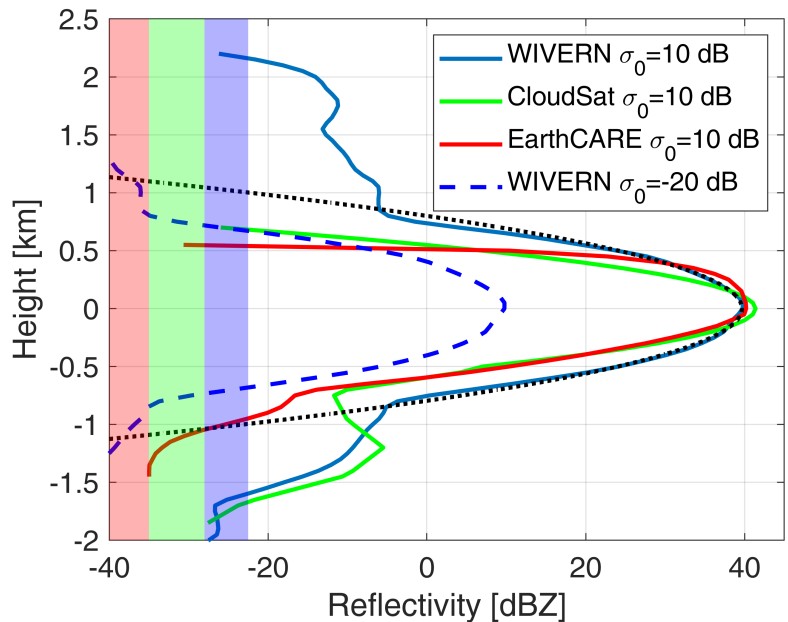

**Figure 2.** Ocean surface clutter profile with $\sigma_0 = 10$ dB for WIVERN (blue), CloudSat (green) and EarthCARE (red). The dotted black profile is the expected WIVERN clutter when computed according to the approximation of Eq. (2). The dashed blue line corresponds to the WIVERN clutter profile with $\sigma_0 = -20$ dB, which accounts for the reduced NRCS of the ocean at WIVERN viewing angle (see Fig. 3). The right limit of the shaded regions corresponds to the minimum detectable reflectivity signal (MDS), driven by the radar receiver noise and the integration length (-35 dBZ for EarthCARE, -28 dBZ for CloudSat and -22.5 dBZ for WIVERN). When the surface clutter profile is higher (lower) than the MDS, the hydrometeor detection of the radar is determined by the clutter (noise) signal.

The shape of the clutter reflectivity profiles for the CloudSat and EarthCARE CPRs are shown in green and red lines in Fig. 2. They have been derived directly averaging ocean surfaces profiles in clear sky conditions. The difference between the CloudSat and EarthCARE profile is mainly driven by different receiver response function that in EarthCARE has been optimized for boundary layer detection (Lamer et al., 2020); it behaves more closely than CloudSat to a top hat function with practically full (no) detection above (below) 500 m. The two clutter profiles are only plotted when above the respective minimum detectable reflectivity signal (MDS). Below this level any atmospheric signal is lost in the noise anyway.




For WIVERN the clutter profile is computed by combining Eq. (1) and Eq. (6), using the WIVERN illumination geometry and its antenna pattern (as provided by industrial studies, ESA-WIVERN-Team (2023)). The result for a top-hat pulse of $3.3\,\mu$s duration is shown in Fig. 2 (blue line) for a surface with $\sigma_0 = 10$ dB, a characteristic value for nadir incidence over the ocean with an 8 m/s wind speed. To a first approximation, the pattern is Gaussian-elliptical, with the antenna exhibiting a narrower 3 dB beamwidth in elevation compared to azimuth (see Tab. 1). The reflectivity profile derived from the Gaussian approximation of the antenna pattern using Eq. (2) is also plotted in the same figure (dotted black line). The difference between the dotted black line (computed with a Gaussian antenna pattern with only a main lobe) and the continuous blue line (computed with the full antenna pattern) demonstrates that for WIVERN the antenna side-lobes significantly broaden the reflectivity profile, resulting in the signal remaining well above the WIVERN minimum detectable reflectivity of -22.5 dBZ for several kilometers above the surface (up to circa 2.2 km).

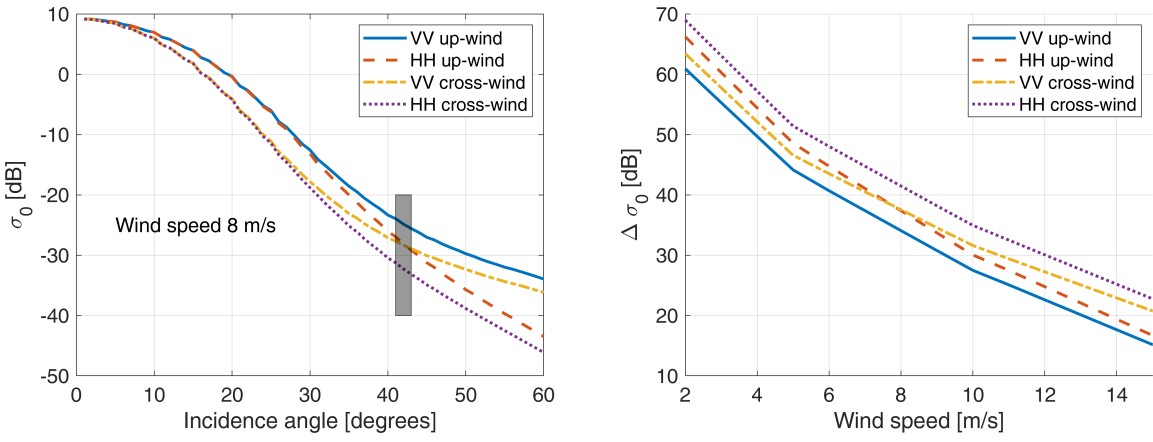

**Figure 3.** Left panel: $\sigma_0$ dependence on the incidence angle for an ocean surface with wind speed of 8 m/s for different polarization and for up-wind and cross-wind observations as indicated in the legend. The geophysical model is based on the work by Battaglia et al. (2017). The grey rectangle corresponds to the value expected for WIVERN. Right panel: the reduction in $\sigma_0$ when moving from nadir to slant observations at $42°$.

In contrast, the reflectivity profiles directly measured for the CloudSat and EarthCARE radars (green and red lines, respectively) are much sharper. The EarthCARE radar, in particular, exhibits a very sharp cut-off at approximately 500 m.

For the ocean surface, there is a significant drop in $\sigma_0$ when transitioning from nadir to slant incidence angles, as demonstrated by spaceborne radar measurements at Ku and Ka bands (Yamamoto et al., 2020) and airborne measurements at W-band (Battaglia et al., 2017). Results from the geophysical $\sigma_0$ model proposed by Battaglia et al. (2017), which account for different polarizations and wind directions, are shown in Fig. 3a for a wind speed of 8 m/s (a characteristic value over the ocean). It is evident that there is a substantial drop relative to nadir, ranging between 20 and 70 dB (Fig. 3b), with the largest reductions occurring under low wind conditions when the ocean surface behaves like a near-perfect mirror. For wind speeds around 8 m/s (the expected mean value over the ocean), drops exceeding 30 dB are anticipated. The simulated WIVERN return for an ocean





surface under such wind conditions corresponds to the dashed blue line in Fig. 2, clearly illustrating the potential of WIVERN
observations in mapping hydrometeors within the boundary layer.

The profiles shown in Fig. 2 are ideal; for CloudSat and EarthCARE real reflectivity signals can be simulated by using a
standard pulse pair processing (Kollias et al., 2014) whereas for WIVERN they can be generated according to the method
proposed by Battaglia et al. (2025) which takes into account the polarisation diversity pulse sequence envisaged for WIVERN
with H and V pairs closely transmitted (with a separation of $20\,\mu s$) and with pairs transmitted every $250\,\mu s$ (Battaglia et al.,
2013). In the following only ideal profiles will be considered with sensitivity levels of -22.5 , -28 and -35 dBZ for WIVERN,
CloudSat and EarthCARE, respectively.

## 3   Forward modeling for radar reflectivity and $SCR$ profiles

Sunsynchronous orbits of the A-Train (local time 2 AM) are used to sample the natural variability of precipitation, water va-
por, temperature, and surface conditions. The Cloud, Aerosol and Precipitation from mulTiple Instruments using a VAriational
TEchnique (CAPTIVATE) algorithm (Mason et al., 2022; Courtier et al., 2024) retrieves microphysical properties (mass content
and characteristic size) of ice, rain, and cloud hydrometeors at a vertical resolution of 60 m and an along-track horizontal reso-
lution of 1.5 km from the CloudSat CPR, the Cloud-Aerosol Lidar with Orthogonal Polarization (CALIOP) and the Moderate
Resolution Imaging Spectroradiometer (MODIS) radiometer observations. Mass contents are provided for each hydrometeor
class; characteristic sizes are available only for ice and rain. Using as input the microphysical properties, the single scatter-
ing properties, specifically effective reflectivities, $z_e^{hydro}(z)$ (in linear units, mm$^6$/m$^3$), and extinction coefficient, $k_{ext}^{hydro}(z)$,
are computed through an interpolation of the profiles with an existing lookup tables dataset. Furthermore, co-located profiles
of temperature, pressure, and relative humidity from the European Centre for Medium-Range Weather Forecasts (ECMWF)
reanalysis are used to compute gas attenuation, $k_{ext}^{gas}(z)$.

Using the aforementioned scattering, absorption and extinction properties of hydrometeors and water vapor, the cumulated
optical thickness from the top of the atmosphere downwards along the WIVERN viewing direction is computed as:

$$\tau(z,\,\theta_{inc}) = \int\limits_{z}^{+\infty} \underbrace{\left[ k_{ext}^{hydro}(z) + k_{ext}^{gas}(z) \right]}_{k_{ext}(z)} \frac{dz}{\cos\theta_{inc}} \tag{10}$$

where a 1D approximation is adopted when considering slant viewing angles (i.e. for each profile the same columnar properties
are assumed everywhere). The NRCS $\sigma_0$ for an ocean surface is computed based on the geophysical model described in
Sect. 2.2 as a function of wind speed, sea surface temperature, and incidence angle. Over land and sea-ice covered surfaces
the $\sigma_0$ value retrieved by CloudSat is used and a constant drop between nadir and the WIVERN incidence angle is imposed.
A decrease in $\sigma_0$ is expected also for land surfaces, but it is highly variable with the surface type (Yamamoto et al., 2020;
Manconi et al., 2024). To account for that this drop, indicated with $\Delta\sigma_0$, is varied between 0 to 20 dB. Then the amplitude of the
surface reflectivity profile, $z_e^{clutter}(z)$, for each of the three radars as described in Sect. 2.2, is adjusted to fit the corresponding
computed $\sigma_0$ value.





Finally the measured reflectivity (measured in $\text{mm}^6/\text{m}^3$) is computed as:

$$z_m^{tot}(z) = \underbrace{\left[ z_e^{hydro}(z) + z_e^{clutter}(z) \right]}_{z_e^{tot}(z)} e^{-2\tau(z,\,\theta_{inc})} \qquad (11)$$

whereas the signal to clutter ratio, $scr$, in linear units, can be defined as:

$$scr(z) = \frac{z_e^{hydro}(z)}{z_e^{clutter}(z)} = \frac{z_m^{hydro}(z)}{z_m^{clutter}(z)} \qquad (12)$$

Reflectivity and signal to clutter ratio are usually measured in dBZ and in dB units (and indicated then with capital letters, $Z$ and $SCR$, respectively) by taking $10 \log_{10}$ of the expression (11) and (12), respectively. Eq. (11) becomes:

$$Z_m^{tot}(z) = Z_e^{tot}(z) - \underbrace{8.686\,\tau(z,\,\theta_{inc})}_{2\,PIA(z,\,\theta_{inc}))} \qquad (13)$$

where the second term is the two way path integrated attenuation (PIA) computed between the top of the atmosphere and level $z$ along the radar line of sight.

The estimation of the $SCR$ profile is the first step in determining if the particular hydrometer profile will be detected by WIVERN. Next, we need to establish a detection threshold value for the $SCR$. Above this threshold value, the hydrometer profile is detected by WIVERN. Given the strong dependency of the $SCR$ with height, the use of a given threshold (e.g., 5 dB) will establish at which height the $SCR$ exceeds a threshold value. For instance $H_{SCR=5\,dB}$ indicates the height at which the $SCR$ will be equal to 5 dB.

### 3.1 Case studies

A stratiform rain precipitation over the Pacific Ocean close to the Western coast of Canada and crossing over the island of Haida Gwaii (Fig. 4) and a snowstorm over the Labrador Sea are used to illustrate the methodology and the results.

#### 3.1.1 Stratiform rain over ocean

A stratiform rain event that occurred off the coast of Canada on 2 January 2008 was chosen to demonstrate the advantages of WIVERN's detection of liquid precipitation on the ground over the ocean. Fig. 5 shows the CAPTIVATE retrieval for ice and rain hydrometeor. Additional information about the total columnar amount of cloud water content and its location is shown in the same figure as a black line and black stars. The freezing level height is between 1.5 and 2 km with a gradual decreasing trend as moving southward. At latitudes below $53°$, rainfall reaches the ground with rates of up to 7 mm/h (black line, bottom panel).

Using the CAPTIVATE products and co-located ECMWF auxiliary data, the W-band scattering properties can be computed as described in Sect.3. Fig.6 illustrates the hydrometeor effective reflectivity and the extinction coefficient (accounting for both hydrometeors and gases), which form the basis for calculating the simulated measured reflectivities. In this example, extinction is particularly high below the freezing level, where liquid hydrometeors are more abundant. Reflectivities in this region exceed 20 dBZ but do not surpass 24 dBZ, due to the well-known saturation effect of reflectivities at the W-band (Hogan et al., 2003).





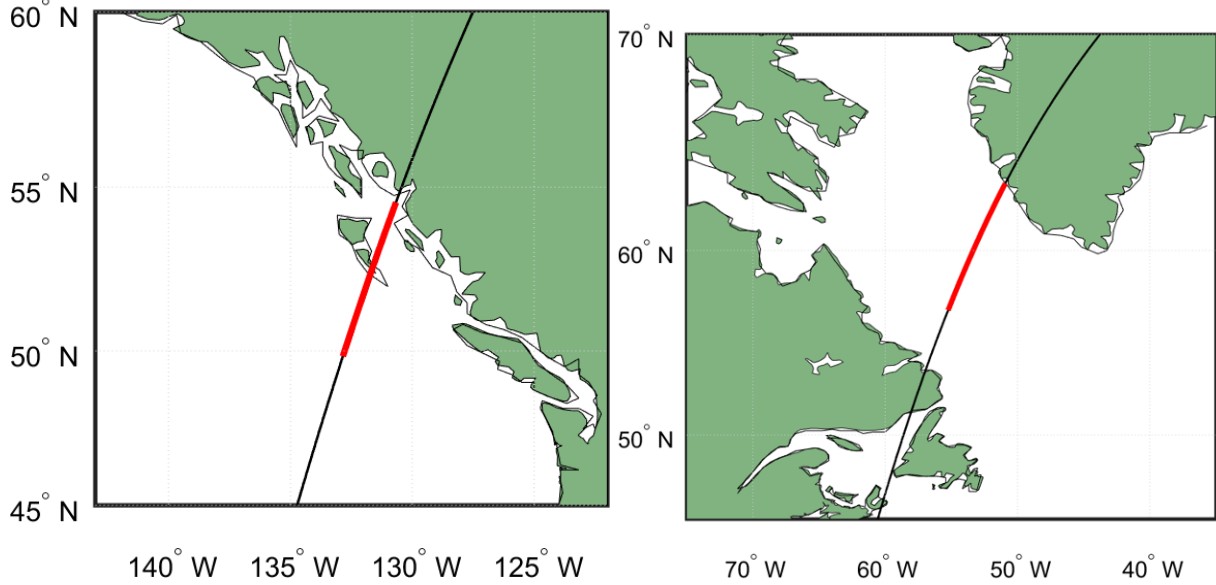

**Figure 4.** Ground-track for the CloudSat orbit in correspondence to the stratiform rain precipitation event off the Western coast of Canada (left panel) and the snowfall event in the Labrador Sea (right panel). The red marked pixels indicate the profiles shown in Fig. 5 and in Fig. 10.

Finally, the key variable in this study is the surface NRCS ($\sigma_0$), which determines the peak value of the clutter profile and thus modulates the signal-to-clutter ratio ($SCR$). Over the ocean, $\sigma_0$ depends on wind speed, sea surface temperature, and the incident angle. Strong winds near the ocean surface ahead of the synoptic-scale precipitation system increase the roughness of the ocean surface, causing a larger WIVERN NRCS ($\sigma_0$) at large incident angles. On the contrary, nadir-looking radars (CloudSat and EarthCARE CPR) observe an increase in $\sigma_0$ under low wind conditions. Furthermore, as shown in Fig. 3, for WIVERN there is a notable difference in the magnitude of $\sigma_0$ between upwind and downwind observations (Battaglia et al., 2017). Surface temperatures also influence $\sigma_0$ by modulating the Fresnel reflection coefficient. Colder temperatures, typically encountered at higher latitudes, result in lower $\sigma_0$ values. Fig. 7 illustrates this $\sigma_0$ behavior by comparing $\sigma_0$ values for nadir observations (CloudSat and EarthCARE) and at a $42°$ incidence angle (WIVERN) for the case study shown in the left panel of Fig. 4.

The clutter profiles are derived by rescaling the profiles shown in Fig. 2 with the computed $\sigma_0$ values. Subsequently, the path-integrated attenuation (PIA) is subtracted from the total effective reflectivity (the sum of hydrometeor and surface clutter contributions) to compute the measured reflectivity (Eq. 13). The simulated measured reflectivities for the three different radars differ due to the path integrated attenuation (oblique vs. vertical, with $PIA_{WIVERN}$ enhanced by a factor of $1/\cos(42°) = 1.35$ compared to $PIA_{CloudSat} = PIA_{EarthCARE}$), the different clutter shapes (Fig. 2) and the different instrument sensitivities.

A comparison between the simulated WIVERN and CloudSat reflectivity profiles is presented in Fig. 8 and demonstrates the following:



**Figure 5.** CAPTIVATE retrieval outputs profiles for a stratiform rain precipitation event that occurred over the Pacific Ocean near the west coast of Canada (see red marked pixels in the left panel of Fig. 4). Top panel: water content with the continuous black line indicating the cloud water path (right y-axis scale) with the star symbols corresponding to the height at which the maximum cloud water content is reached. Bottom panel: median volume diameter with the continuous black line corresponding to rainfall rate (right y-axis scale). The left (right) colorbar relates to the rain (ice) characteristics.





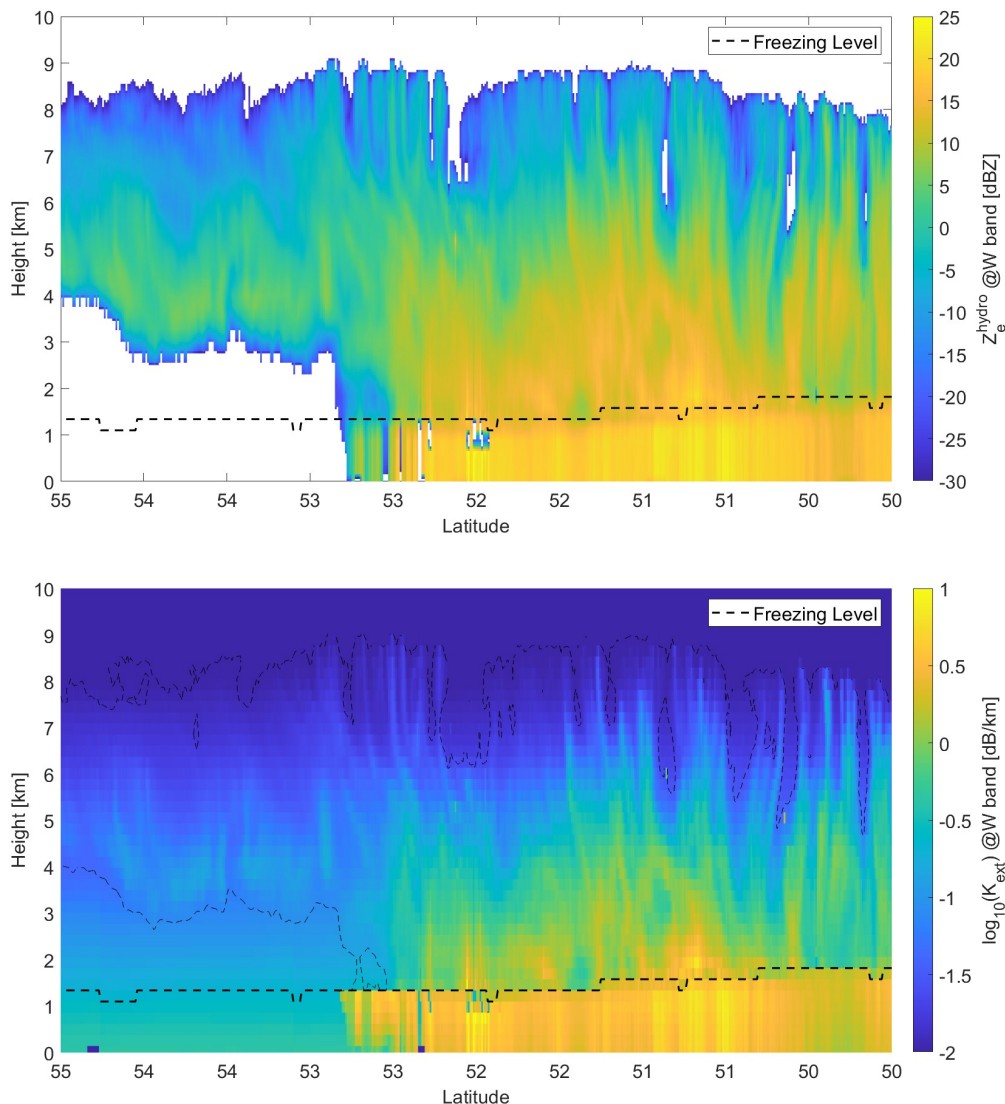

**Figure 6.** W-band vertical profiles of effective reflectivity (top) and W-band hydrometeor extinction coefficient (bottom) in correspondence to the microphysical properties shown in Fig. 5. The dashed black line corresponds to the freezing level.

– The reduced sensitivity of WIVERN, as cloud top edges are not detected by the radar because they fall below the sensitivity threshold of -22.5 dBZ.

– Increased attenuation due to the slant view, especially at lower levels, where attenuation is stronger below the freezing level because of the presence of water hydrometeors and higher concentrations of water vapor.

– A reduced clutter signal in the WIVERN reflectivity over the ocean (in regions without gray shading).





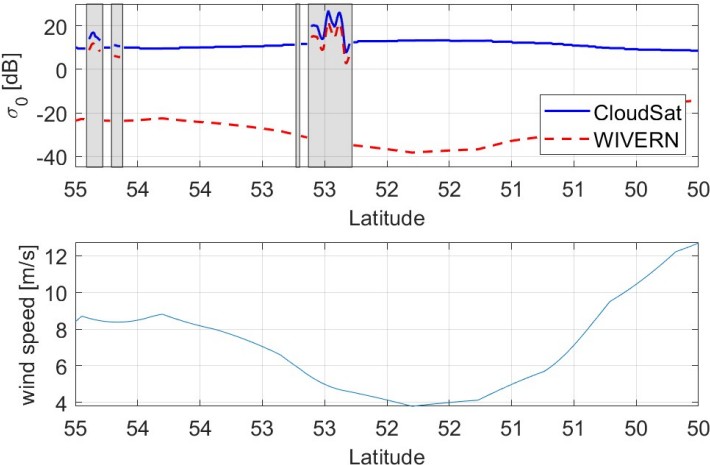

**Figure 7.** Top panel: simulated $\sigma_0$ for nadir looking radar like CloudSat (blue line) and for a $42°$ incidence angles (WIVERN)(red dashed line). Note that over land a $\Delta\sigma_0 = 5$ dB has been assumed. Bottom panel: wind speed to show the dependence of WIVERN $\sigma_0$ from it. Shaded areas correspond to land where the $\sigma_0$ is assumed to be independent of the incidence angle.

   – A thicker clutter signal in the WIVERN reflectivity when flying over land (gray-shaded regions).

   The height at which the signal-to-clutter ratio (SCR) equals 5 dB is also plotted in the figure as a continuous black line. The 5 dB threshold has been selected to identify hydrometeor signals that are not contaminated by surface clutter. Notably, in the central part of the segment where rainfall is present (latitudes between $50°$ and $52°$), where winds are weaker, WIVERN
demonstrates a clear advantage over CloudSat in detecting atmospheric targets close to the ground and retrieving precipitation reaching the surface.

   The vertical profile corresponding to the dashed red line in Fig. 8 is analyzed in detail in a separate panel (Fig. 9). This analysis highlights the different clutter shapes for the three radars (compare the three dashed curves), which result from differences in illumination geometry (nadir versus slant) and receiver response functions (CloudSat versus EarthCARE). EarthCARE and
CloudSat show the same hydrometeor-attenuated reflectivity profile (star symbols), while WIVERN (blue stars) exhibits lower reflectivities due to enhanced path-integrated attenuation. In the right panel of Fig. 9, the SCR plot clearly demonstrates the substantial improvement achieved in the WIVERN configuration. The three circles, representing the height at which the SCR equals 5 dB, demonstrate that such level is practically at ground level for WIVERN, whereas it is approximately 500 m above the ground for EarthCARE and CloudSat. Also note that CloudSat has in fact a better SCR than EarthCARE in the lowest
500 m.

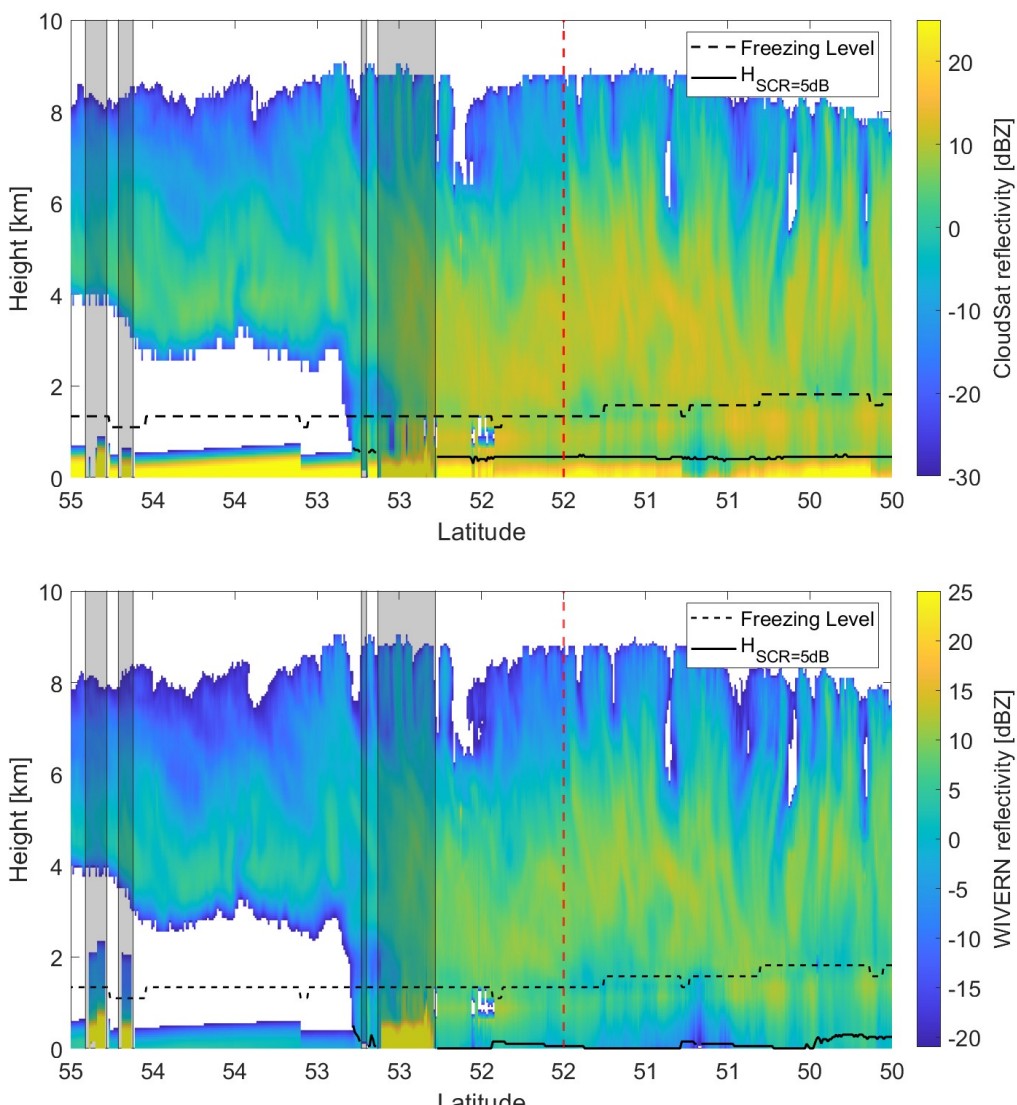

**Figure 8.** Simulated WIVERN (top) and CloudSat (bottom) measured reflectivity profiles. The dashed cyan line corresponds to the freezing level whereas the black continuous line corresponds to $H_{SCR=5\,dB}$. The region which is partially shaded corresponds to land.



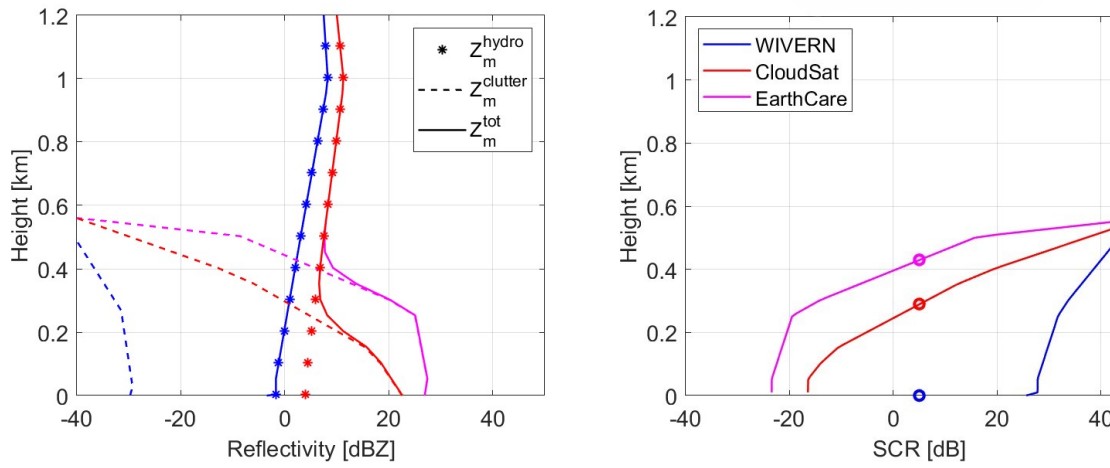

**Figure 9.** Hydrometeor, clutter and total reflectivity (left panel) and SCR (right panel) for the three radar configurations of this study (blue for WIVERN, red for CloudSat and magenta for EarthCARE) for the profile corresponding to the vertical dashed red line in Fig. 8. The circles correspond to the level where SCR is equal to 5 dB; for WIVERN $SCR > 20$ dB at the surface. In this case a value of 0 km is assigned to the variable $H_{SCR=5\ dB}$.

### 3.1.2 Convective snowfall over ocean case study

Another important application of WIVERN improved ground precipitation detection is for oceanic snowfall at high latitudes. Snow occurs in deep stratiform systems and shallow events with cloud tops lower than 2000 m, often associated with cold-air outbreaks, which can significantly contribute to the total annual accumulation (e.g. Kulie et al. (2016); Kulie and Milani (2018);
Battaglia and Panegrossi (2020)).

The selected case study shows two distinct convection cells over the Labrador Sea (right panel of Fig. 4) with the micro- and macro-physical structures of the system shown in the top panels of Fig. 10. Here, the difference between the CloudSat and WIVERN $\sigma_0$ is large ($\approx 55 - 60\ dB$) due to the very low wind near-surface speeds that smooth the sea surface creating a great return for nadir-looking radars, but a very weak one for WIVERN. This results in a very weak surface clutter for WIVERN,
ideal for near-surface retrievals, and a very strong one for CloudSat, almost obscuring the 500 m near the ground (see bottom panels in Fig. 11).

The snow convective cores with the high ice water content result in high snowfall rate (bottom left panel of Fig. 10). Often these areas are characterized by updrafts aloft and the presence of thick supercooled layers (top left panel of Fig. 10). These supercooled liquid layers can cause large attenuation (top right panel of Fig. 11) which can be sufficient to drive the signal
below the WIVERN detectability threshold. Thus, in contrast to what we have concluded with the rain event, a milder snowfall will be better detected down to the ground than a heavier one because the signal may drop below the sensitivity threshold.





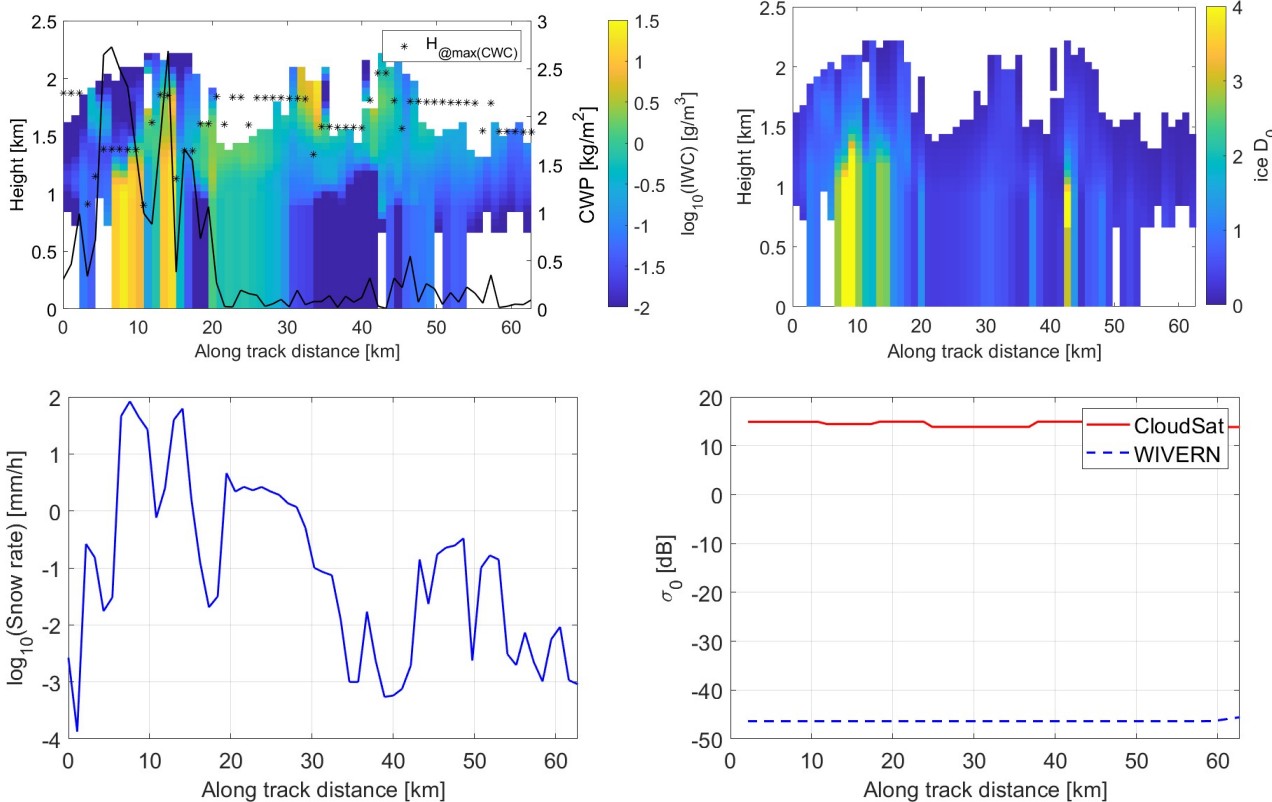

**Figure 10.** Snow precipitation event occurred over the Labrador Sea (see red marked pixels in the right of Fig. 4). The top panels show the CAPTIVATE retrieval profiles as a function of the along track distance. Left panel: ice water content and cloud water path (black line) with stars corresponding to the height at which the largest cloud water content is present. Right panel: snow median volume diameter). The bottom left panel depicts the snowfall rate (in logarithmic units) whereas the bottom right panel presents the $\sigma_0$ for CloudSat and WIVERN (red and blue lines, respectively).

## 3.2 Statistical analysis

In addition to the two case studies presented in detail, a database of precipitation cases has been constructed using a total of 1,200 A-Train orbits, encompassing a wide range of different conditions. The ocean rainfall profiles are selected using the CloudSat 2B-PRECIP-COLUMN product. The snowfall profiles are grouped into two categories: those observed over an ice-free ocean surface and those observed over land and sea-ice conditions. All profiles are subsequently clustered according to the mean value of $Z_e^{hydro}$ in the first km above the ground ($\langle Z_e^{hydro} \rangle_{[0-1]\,km}$) in reflectivity classes ranging from -15 to 25 dBZ with 5 dB width.

Histograms of the minimum height $H_{SCR=5\,dB}$ are generated for each class and the corresponding mean, percentiles, and standard deviations are subsequently calculated from such histograms. The results are summarized in the form of box plots for



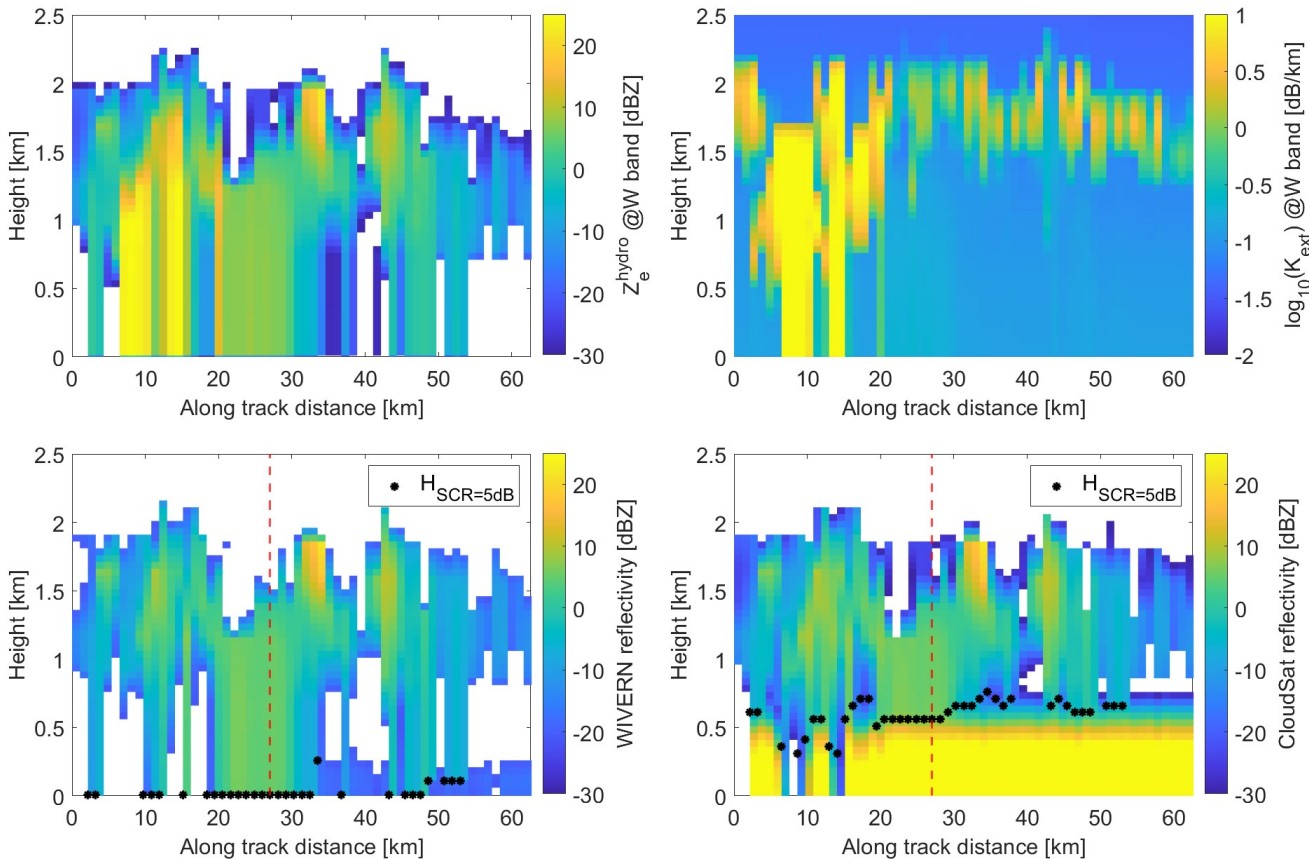

**Figure 11.** Reconstruction of WIVERN and CloudSat observations for the oceanic snowfall event whose ground track is shown in the right panel of Fig. 4. Top panels: hydrometeor effective reflectivity (left) and extinction coefficient (right) at W band. Bottom panels: WIVERN (left) and CloudSat (right) simulated measured reflectivity.

the three different configurations (rainfall over ocean, snowfall over ice-free ocean, snowfall over land or sea-ice) in Fig. 13. The results clearly highlight that WIVERN would have a significant advantage in terms of improved $SCR$ compared to nadir-looking radar over ocean surfaces for both liquid and solid precipitation, especially for moderate precipitation rates. For a threshold of 5 dB in SCR, the lowest height WIVERN can detect precipitation over the ocean improves by 300 m to 400 m compared to CloudSat and EarthCARE On the other hand, over land or sea-ice WIVERN is performing worse than CloudSat and EarthCARE in the hypothesis that $\Delta\sigma_0 = 5$ dB but it is practically equivalent to the two other systems if $\Delta\sigma_0 = 20$ dB. In addition, WIVERN will have reduced detectability due to lower sensitivity particularly in areas of strong attenuation ($\langle Z_e^{hydro}\rangle_{[0-1]\,km} > 15$ dBZ) and of weak signal ($\langle Z_e^{hydro}\rangle_{[0-1]\,km} < -15$ dBZ) as shown in Fig. 14.

Despite these limitations over land, thanks to its much better sampling (Scarsi et al., 2024), WIVERN is still expected to significantly improve our understanding of snowfall climatology at regional and seasonal scales over all types of surfaces.



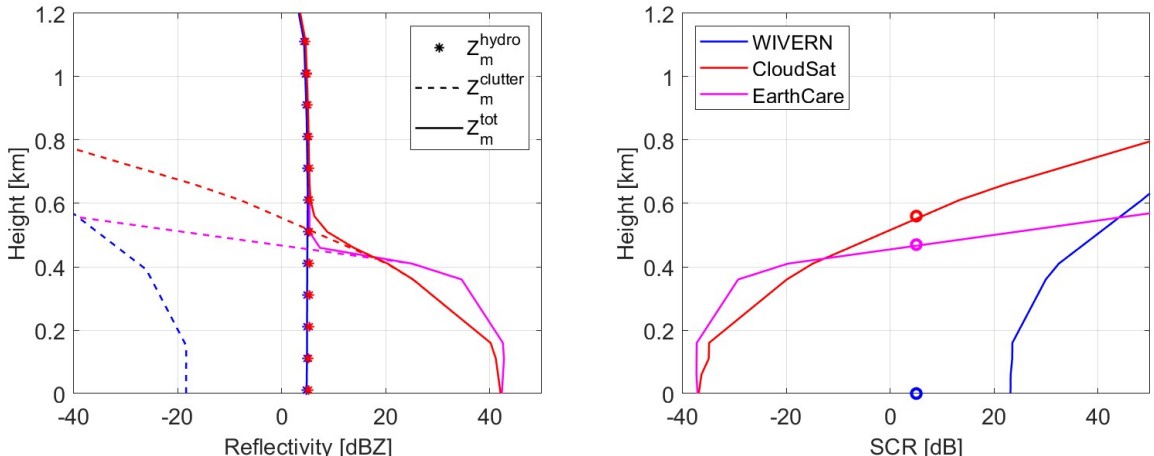

**Figure 12.** Hydrometeor, surface clutter and total measured reflectivity (left panel) and SCR (right panel) for the three radar configurations of this study (blue for WIVERN, red for CloudSat and magenta for EarthCARE) for the profile corresponding to the vertical dashed red line in Fig. 11.

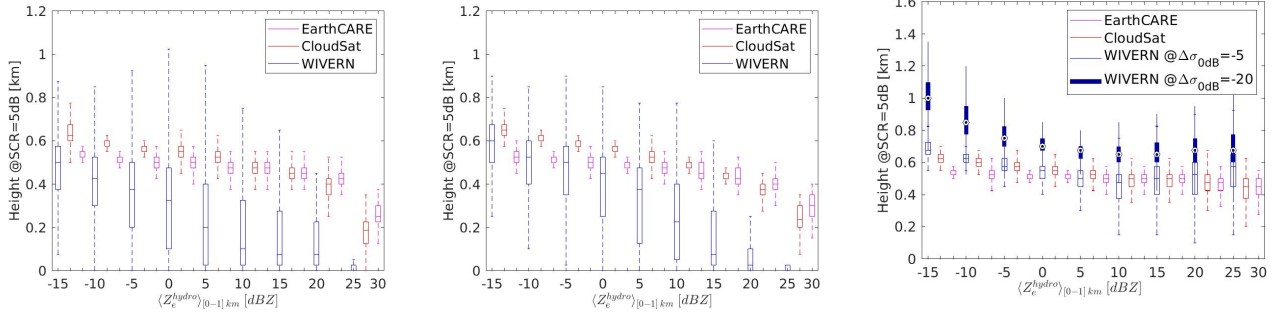

**Figure 13.** Box plots of the distributions of $H_{SCR=5\,dB}$ as a function of the mean value of $\langle Z_e^{hydro}\rangle_{[0-1]\,km}$ for oceanic rainfall (left), snowfall (center) and snowfall over land or sea-ice (right). In the box plots the central mark indicates the median and the bottom and top edges of the box indicate the 25th and 75th percentiles, respectively whereas the whiskers extend to the most extreme data points not considered outliers. In the right panel, there are two box plots for WIVERN corresponding to a $\Delta\sigma_0 = -5$ dB and 20 dB, respectively.

### 3.3 Impact on reflectivity measurements

In order to better quantify the gain associated to the reduced clutter height, the probability distribution functions of the difference between the hydrometeor effective reflectivity near the ground and at $H_{SCR=5\,dB}$ for the different configurations have been computed. Since there is not high confidence that CAPTIVATE inverted profiles near the ground really capture the vertical variability of precipitation, two datasets of reflectivity gathered from ground-based sites have been used.



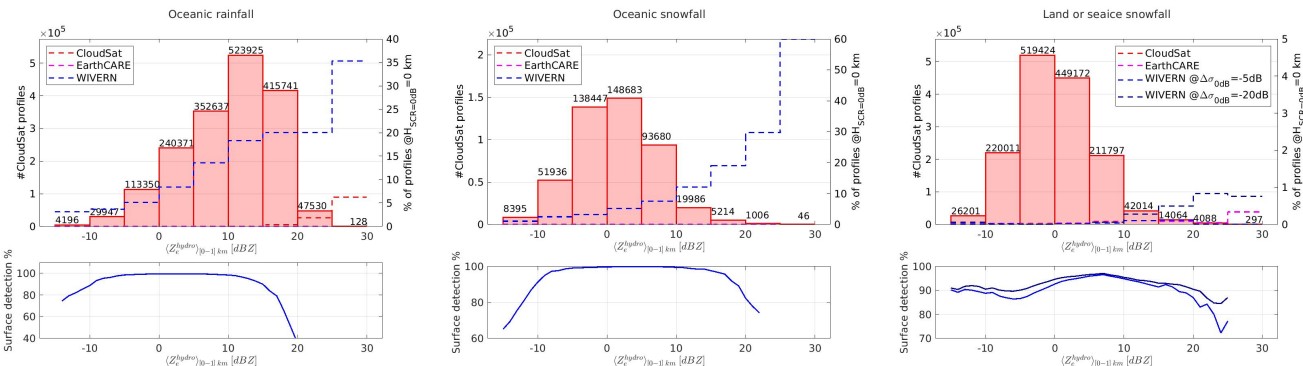

**Figure 14.** Top: Number of CloudSat profiles selected for statistical analysis, for oceanic rainfall on the left, oceanic snowfall in the middle panel, and land or sea ice snowfall on the right; profiles are clustered by effective reflectivity in surface classes from -15dBZ to 30dBZ, each 5dBZ wide. Dashed lines are the percentage of profiles detected at ground level ($H_{@SCR=5dB} = 0km$) and are shown for WIVERN (blue), CloudSat (red) and EarthCARE (magenta). Bottom: Proportion of profiles detected by WIVERN compared to those detected by CloudSat as a function of $\langle Z_e^{hydro} \rangle_{[0-1]\,km}$. The profiles detected by Cloudsat are the 100% baseline. Results are only shown for reflectivity classes with more than 100 profiles.

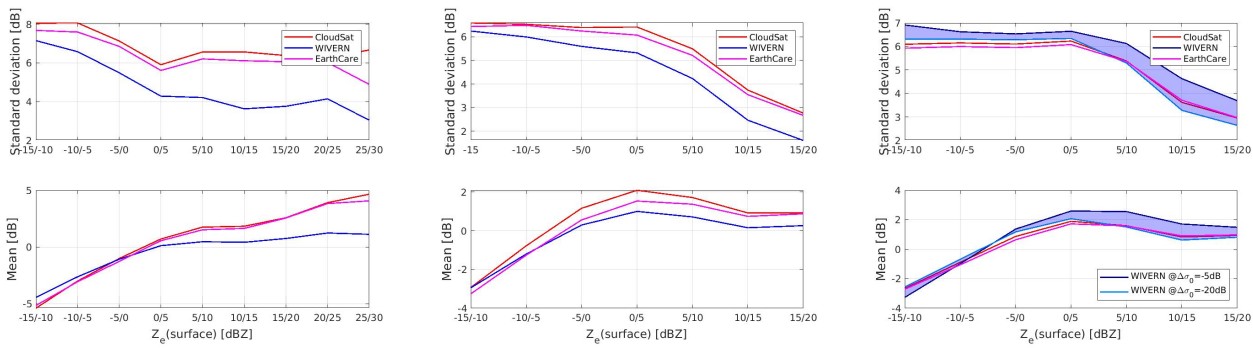

**Figure 15.** Standard deviation (top row) and mean values (bottom row) for the pdfs of $\Delta Z$ as a function of surface reflectivity for oceanic rainfall (left), snowfall (centre) and land snowfall (right) for CloudSat (red), EarthCare (magenta) and WIVERN (blue). For land snowfall, the shaded region corresponds to the results obtained from the pdfs of $\Delta Z$ for WIVERN with a $\Delta\sigma_0$ ranging from 5 to 20 dB.

### 3.3.1 Snowfall

The first site is located on the northern slope of Alaska (71°N, 156°W) approximately at sea level and the ground radar equipment is the Ka ARM Zenith Radar (KAZR) (Widener et al., 2012). Vertical profile observations at 30 m resolution are averaged every minute for a total of 5 months from November 2011 to April 2011. Only timestamps with subfreezing 265 temperatures at the ground are down-selected. Profiles of Ka-band measured reflectivities are converted to profiles of effective



W-band reflectivities by adopting the transfer function proposed by Kollias et al. (2019). Attenuation due to snow, supercooled clouds or atmospheric gases at Ka is neglected.

The W-band reflectivity profiles obtained by this procedure are used to compare the effective reflectivity at the ground level with that at the level where the SCR is equal to 5 dB. The different probability density functions of $\Delta Z = Z_e(H_{SCR=5dB}) - Z_e(surface)$ for the three different configurations are computed by using the distributions of $H_{SCR=5dB}$ depicted in the center and right panels of Fig. 13. The mean and the standard deviation computed from these pdfs are shown in Fig. 15. For snowfall over ocean (left panels) WIVERN clearly shows an advantage compared to EarthCARE and CloudSat with a standard deviation which is 1 dB better for the class of reflectivity between -5 and 0 dBZ and steadily increases to 2 dB better at large surface reflectivities. The bias is also reduced. The situation reverses when considering snowfall over land or sea-ice but only for the worst case scenario for WIVERN of $\Delta\sigma_0 = 5$ dB. In the more optimistic scenario with $\Delta\sigma_0 = 20$ dB WIVERN, CloudSat and EarthCARE performs very similarly apart for a deterioration of WIVERN performances at high values of $Z_e(surface)$.

### 3.3.2 Rainfall

The second dataset is extracted from W-band observations from the Barbados Cloud Observatory (BCO) (Stevens et al. (2016); Lamer et al. (2015)). A total of 21 months of observations during the period 2018-2021 when the W-band radar was operational are used to characterize shallow precipitation in the tropical oceans. The attenuated reflectivity profiles are corrected for attenuation with an iterative correction based on the technique proposed by Hitschfeld and Bordan (1954) to compute the effective reflectivity profiles. A quadratic relationship between the $\log_{10}$ of the W-band extinction coefficient in dB/km and the radar reflectivity in dBZ has been assumed:

$$\log_{10}(k_{ext}^{rain}) = -0.437 + 0.013\, Z_e + 0.0017\, Z_e^2 \tag{14}$$

The correction is deemed appropriate because very high rainfall rates are excluded from this dataset so that total attenuations in the 2 km closest to the surface rarely exceed 10 dB. Similarly to the procedure followed for oceanic snowfall events, the statistical analysis of $\Delta Z$ is conducted, investigating the mean and the standard deviation of the distributions for different surface effective reflectivity classes (left column in Fig. 15). Results demonstrate that WIVERN will outperform EarthCARE and CloudSat both in terms of biases and standard deviations for all surface reflectivities ranging from -10 to 25 dBZ.

## 4 Summary and conclusions

The WIVERN conically scanning Doppler W-band radar, currently undergoing Phase A studies within the competitive EE11 program, could usher in a new era of spaceborne cloud radars. It has the potential, for the first time, to map the mesoscale and synoptic variability of horizontal winds, cloud dynamics, and precipitation microphysics on a global scale. One of the key features of the WIVERN 94-GHz radar is its reduced Earth's surface reflection. This study shows that the oblique angle of incidence (approximately 42°) will be advantageous compared to standard nadir-looking radars due to substantial clutter suppression over ocean surfaces, thanks to the large drop in surface normalized radar cross-section. This feature will enable





the detection and quantification of light and moderate precipitation (both liquid and solid phases) over the ice-free ocean, with improved proximity to the surface compared to what has been achieved by CloudSat and currently by the EarthCARE CPR.

For snow precipitation over land or sea-ice, WIVERN clutter contamination is expected to slightly degrade precipitation estimates if the drop in normalized radar cross-section from nadir to the WIVERN viewing slant direction is marginal (5 dB), or perform nearly as well as EarthCARE and CloudSat if this drop is substantial (20 dB). Currently, a thorough characterization of $\sigma_0$ variability at 94 GHz with incidence angle for ice- and snow-covered surfaces is lacking; future airborne or ground-based campaign should produce detailed $\sigma_0$ characterization for the 94 GHz frequency and at very slant incidence angles..

Light and moderate precipitation, which is a critical component of the water cycle in high-latitude oceans, remains poorly mapped by the current global observing system, with uncertainties still on the order of tenths of a millimeter per day (Petković et al. (2023), their Fig. 10). Inconsistencies between satellite precipitation products in retrieving light rain and the limitations of the current observing system are the main contributors to this uncertainty (Battaglia et al., 2020; Schulte et al., 2022, 2023).

Future work should focus on developing and assessing retrieval algorithms that fully leverage WIVERN observations, including reflectivities and polarized brightness temperature, to improve rainfall and snowfall estimations.

*Code and data availability.* Simulator code and all raw data used are available on request.

*Author contributions.* MC performed most of the simulations and the analyses. AB contributed to the analysis, the writing and has defined the project. PK and FT contributed to the review of the paper.

*Competing interests.* At least one of the (co-)authors is a member of the editorial board of Atmospheric Measurement Techniques.

*Acknowledgements.* This research has been supported by the European Space Agency under the activities "WInd VElocity Radar Nephoscope (WIVERN) Phase A Science and Requirements Consolidation Study" (ESA Contract Number RFP/3-18420/24/NL/IB/ab) and "End-to-End Performance Simulator Activity of the WIVERN Mission" (ESA Contract Number 4000139446/22/NL/SD). This research used the Mafalda cluster at Politecnico di Torino. The work by MC has been funded by the Space It Up project funded by the Italian Space Agency, ASI, and the Ministry of University and Research, MUR, under contract n. 2024-5-E.0 - CUP n. I53D24000060005.



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
