# Peer review of "Improved hydrometeor detection near the Earth's surface by a conically scanning spaceborne W-band radar"

_EGUsphere, 2025_

## Author Response (AR1)

**Response to referee comment #1**

**-   There are a number of other WIVERN publications that are not cited.  It would be good to confirm that all clutter-related papers are referenced and that this paper explains its novelty relative to those.**

All the relevant papers to the clutter study have been cited all throughout the text. We add a sentence in the intro with reference to the paper by Manconi et al. 2025 and have changed the intro to make more clear the originality of the paper.

**-   Around line 50, the advantages of conically-scanned W-band radar are noted, but it might be reasonable to list some disadvantages, in particular the large attenuation that needs to be removed when viewing liquid precipitation.**

A statement has been added to the paper. ("On the other hand the conical scan complicates the interpretation of the measurements, increases the attenuation due to the longer slant path and the impact of antenna side-lobes").

**-   I think the equations developed in Section 2.1 need better motivation, especially the use of the Gaussian antenna.  It appears that the WIVERN calculations use a calculated pattern.  Is the Gaussian used for CloudSat and EarthCARE? Is (7) the clutter reflectivity being plotted in Figure 2?**

Gaussian antenna describes very well the behaviour of the main lobe even for WIVERN (compare black dotted and blue line in Fig.2). since all expressions are analytical we thought it was useful to report them. We introduce a new sentence to motivate our choice.

**-   Do the results in Figure 2 represent the total clutter, from all pulses that could arrive simultaneous with the atmospheric pulse, or only from the pulse illuminating the atmosphere?  If only the latter, I would be concerned that the clutter estimate is not complete.**

We have neglected this effect. If we understand correctly the reviewer's comment he is concerned for the next and following pulses potentially interacting with the main pulse. With the PRF envisaged for WIVERN (3kHz) and the geometry of WIVERN for the closest pulses to get the same range as the main pulse the angular displacement should be of the order of 4-5° compared to the boresight. Since the antenna gain at such angles is more than 40 dB lower than the gain at the boresight we expect this will not be a problem.

**-   Around line 125, it could be pointed out that even at the 14 m/s windspeed, the clutter appears small enough that it would be below the thermal noise for WIVERN.**

Not sure we understand the reviewer's comment. As shown in Fig.2 the clutter is not below

WIVERN thermal noise. To be so the sigma0 need to drop roughly 60 dB compared to nadir (this may occur exceptionally at very low winds in the doldrums).

**- Line 193, the text could mention the land/ocean contrast that can be seen in Figure 7.**

The text will be modified according to the comment.

**- Line 206: The text points out a thicker clutter signal when flying over land.  This is related to the total clutter level and its profile, as well as the color table being used for the image.  It might be clearer to mention the level and the 3-dB width.**

The text will be modified according to the comment, both figures will be modified adding the level of the SCR=5dB in the land as well.

**- Figure 8 compares the simulated WIVERN and CloudSat data.  What are the vertical resolutions (not slant range) in the two cases?**

Both platform simulations have a vertical resolution of 140m.

**- The smaller figure format used for the Labrador case might also work for the west coast case.  The larger figures for that case are easier to read but also require a lot of moving around to see both text and figures, when reading.**

The figures will be adapted to the required format

**- I think the text for the snowfall case could be expanded to give more explanation for what is seen.  Is the main difference (better detection down to the ground for snowfall) related to the lack of strong attenuation?**

The text will be modified as follows to clear out the difference: Ice scattering properties in the W band ensure low attenuation, when only solid phase clouds are present, and thanks to the slant view over sea water the contribution given by the surface clutter is minimal.

**- The statistical analysis in 3.2 could also provide a better explanation.  In particular, line 247 notes that the "results clearly highlight".  It would be good to point out the particular results in detail, explaining the choice of the definition of DeltaZ and explaining the significance of the plots.  This explanation would also help with understanding of the results in Section 3.3.  What was the reason for using the SCR=5 dB height?**

The text will be modified according to the commentThe 5dB threshold has been selected on an arbitrary basis from a range of values for which the signal is not contaminated by surface clutter. It is important to note that, given that this is a comparative analysis, the specific threshold selected is inconsequential, provided that it is applied uniformly to both cases.

**- Line 255 – in what sense is the WIVERN sampling improved?  Is this related to the wide swath, which is, indeed, a nice feature?**

WIVERN has improved sampling capabilities compared to nadir pointing radars due to its conically scanning radar that provide many more samplings and coverage.

**Minor edits:**

**- Line 27, the parentheses make reading the sentence difficult. Please consider revising.**

The text will be adapted to the comment

**- Consider removing paragraph breaks at lines 32, 45, 57, 117, 308.**

The text will be adapted to the comment

**- Figure 1, please add the angles used in Section 2.1.**

The figure will be adapted to the comment

**Response to referee comment #2**

- **In general, one could emphasise in the introduction that this refers to reduced clutter over oceans rather than over land. For non-experts, this might give the impression that an increased detection of kidney strike is generally expected worldwide. However, that is not the case, is it?**

  A sentence has been introduced at the beginning to clarify the distinction between ocean and land.

- **I suggest adding a flowchart or a sketch that illustrates all the angles and areas mentioned in the introduction of the formulas. Alternatively, could you add this information to Figure 1? In Equation 1, the incidence angle is yindicated, but in Figure, is S from Equation 1 the darker blue circle in Figure 1?**

  The figure has been updated

- **Figure 2: Why is the black dotted line not in the legend?**

  Its description is present in the caption. The curve has been introduced for showing how good is the Gaussian approximation. It is not in the caption to avoid cluttering too much the figure (the dotted curve is not the main message of the figure).

- **The text would generally benefit from more precise labelling of the figures. Particularly in the case study and statistical analysis section, many small and large plots are combined into a single figure. Numbering them with a), b), etc., would greatly enhance the text's readability. This approach allows figures to be recognised more quickly, helping the reader locate the correct figure faster.**

The text will be adapted to the comment

- **Section 3, line 141: Which lookup tables were used precisely? Are there any references? Likewise, for the calculation of the gas absorption (line 143)**

The lookup tables are computed using Mie and Rayleigh-Gans models for single particle scattering for each hydrometeor (details in Sect 2.22. in Battaglia, A., Martire, P., Caubet, E., Phalippou, L., Stesina, F., Kollias, P., and Illingworth, A.: Observation error analysis for the WInd VElocity Radar Nephoscope W-band Doppler conically scanning spaceborne radar via end-to-end simulations, Atmos. Meas. Tech., 15, 3011–3030, https://doi.org/10.5194/amt-15-3011-2022, 2022.). Variations of Temperature, mean mass weighted diameter and mu parameter have been accounted for in the lookup tables.

**Section 3 includes numerous illustrations. To enhance information extraction from them, the text requires additional explanations or more detailed descriptions of the figures, making it easier for readers to understand and access the information the writer intends to convey. For non-experts, the statistical information is otherwise quite challenging to grasp. A suggestion could be to summarise the data and reduce the number of illustrations to help manage the overwhelming amount of information.**

The text will be adapted to the comment

---

## Author Response (AR2)

**Response to referee comment #1**

**- Around line 55, I noticed the explanation of the increase in the signal/clutter ratio. It sounds like the clutter varies by lambda^4, but it's the particle reflectivity. Consider re-phrasing to clarify.**

Rephrased as follows: the smaller wavelength (W vs Ka and Ku), which favors larger signal-to-clutter ratios due to the different wavelength dependence of surface versus hydrometeor reflectivity

**- Line 79, the description of theta_b being the along range beamwidth isn't clear to me. When I looked up Kanemaru's description, it is "the −6-dB width of the two-way antenna pattern in the cross-track direction." This seems clearer.**

It was indeed an error, corrected as suggested.

**- In section 2.1, I think it's clear now that the Gaussian is used for all radars. I had originally understood from what's now line 112, that the full pattern was used for WIVERN. Apparently, it's only the beamwidth and gain for the Gaussian model. To avoid confusion, the use of the patterns mentioned in line 112 could be explicitly stated.**

Phrase added to clarify that we, in the paper, are using for the simulation the full antenna pattern.

**- In trying to decipher my original comment about Figure 2 and clutter at the 14 m/s wind, it appears that I misread something. Your comment is that "To be so the sigma0 need to drop roughly 60 dB…" Figure 2 shows the case of a 40 dBZ at nadir versus the -20ish thermal noise. Hence, to get to the thermal noise level, the clutter needs to be suppressed by the 60 dB you mention in your reponse, correct?**

Yes, a phrase to clarify the point has been added: Even in these extreme conditions the surface signal would still be above the thermal noise because the $\sigma 0$ in nadir conditions will be approximately 20 dB so that the WIVERN surface profile will look like the blue curve in Fig. 3 reduced by approximately 50 dB = 60−(20−10) dB.

**- The figure captions need revision to match the layout, e.g. Figure 6 caption say top/bottom versus figures that are left/right. Same with Fig. 7, except the caption is right/left but panels are top and bottom.**
Captions have been adjusted accordingly.

**-    The choice of colors for CloudSat and EarthCARE make them difficult to distinguish in some plots, e.g., Figure 13.**

Colors have been adjusted accordingly.

**-    In re-reading the text in Section 3.3 and examining Fig. 15, I wanted to verify that delta-Z is only hydrometeors; in other place Z_e for the hydrometeors uses a superscript "hydro".**

Yes DeltaZ is referring to difference in effective reflectivities of hydrometeors, the symbols have been adjusted accordingly.